# An Oval-Square Shaped Split Ring Resonator Based Left-Handed Metamaterial for Satellite Communications and Radar Applications

**DOI:** 10.3390/mi13040578

**Published:** 2022-04-07

**Authors:** Ismatul Nisak Idrus, Mohammad Rashed Iqbal Faruque, Sabirin Abdullah, Mayeen Uddin Khandaker, Nissren Tamam, Abdelmoneim Sulieman

**Affiliations:** 1Space Science Centre (ANGKASA), Institute of Climate Change (IPI), Universiti Kebangsaan Malaysia, Bangi 43600, Malaysia; ismatulnisak.idrus@gmail.com (I.N.I.); dr_sabirin@ukm.edu.my (S.A.); 2Centre for Applied Physics and Radiation Technologies, School of Engineering and Technology, Sunway University, Bandar Sunway 47500, Malaysia; mayeenk@sunway.edu.my; 3Department of Physics, College of Science, Princess Nourah bint Abdulrahman University, Riyadh 11671, Saudi Arabia; nmtamam@pnu.edu.sa; 4Department of Radiaology and Medical Imaging, Prince Sattam Bin Abdul Aziz University, Alkharj 11942, Saudi Arabia; a.sulieman@psau.edu.sa

**Keywords:** C-band, left-handed metamaterial, oval-square, satellite, X-band

## Abstract

Development of satellite and radar applications has been continuously studied to reach the demand in the recent communication technology. In this study, a new oval-square-shaped split-ring resonator with left-handed metamaterial properties was developed for C-band and X-band applications. The proposed metamaterial was fabricated on 9 × 9 × 0.508 mm^3^ size of Rogers RO4003C substrate. The proposed metamaterial structure was designed and simulated using Computer Simulation Technique (CST) Microwave Studio with the frequency ranging between 0 to 12 GHz. The simulated result of the proposed design indicated dual resonance frequency at 5.52 GHz (C-band) and 8.81 GHz (X-band). Meanwhile, the experimental result of the proposed design demonstrated dual resonance frequency at 5.53 GHz (C-band) and 8.31 GHz (X-band). Therefore, with a slight difference in the dual resonance frequency, the simulated result corresponded to the experimental result. Additionally, the proposed design exhibited the ideal properties of electromagnetic which is left-handed metamaterial (LHM) behavior. Hence, the metamaterial structure is highly recommended for satellite and radar applications.

## 1. Introduction

Numerous studies have been conducted to enhance technology, especially in microwave applications. Over the past 20 years, scientists have introduced metamaterials that have unusual properties that are unattainable in conventional materials. Metamaterial applications have grown in popularity in recent technology research due to their unique interaction with electromagnetic waves [1]. According to the statistics determined by Askari et al., in 2020, the number of papers published from 2000 to 2019 in the metamaterial field shows the emergence of metamaterial research [2]. The idea of introducing metamaterials is mainly to produce materials using artificially engineered and fabricated structural units to achieve the desired unique properties and functionalities.

Veselago first discovered unique properties such as reversed Cherenkov radiation, in which electromagnetic radiation was emitted in the opposite direction of the moving particle charge in left-handed material (LHM) [3]. Veselago also predicted that LHM would exhibit unusual characteristics such as a negative refractive index and a reversed Doppler effect. The structural unit materials can be designed in different shapes and sizes such that the refractive index of the metamaterial will turn into positive, near-zero, or negative values. Therefore, different properties of metamaterials can be classified into double-positive materials (DPS), single-negative materials (SNG), and double-negative materials (DNG) [4]. Conventional materials or materials that exist in nature mostly have positive values of both permittivity and permeability. These materials are known as DPS materials. Materials with a negative value of either permittivity or permeability are called SNG materials. If the permittivity value is negative, the materials are named ENG (epsilon negative), whereas the materials are called MNG when the permeability is negative. Artificial materials that exhibit negative values for refractive index, permittivity, and permeability are known as DNG and can also be called left-handed materials (LHM). The expansion of metamaterial research, which revealed unique capabilities of the metamaterial itself, has resulted in a wide range of applications such as satellite [5], radar [6], perfect absorbers [7], energy harvesting [8], protection of buildings during earthquakes [9], superlenses [10], antenna [11], cloaking [12], sensor [13], and other applications in microwave and optical frequencies.

The C-band and X-band are the microwave frequency bands for satellite communications and radar applications. Satellites and radar, in general, have a wide range of applications, including communications, navigation, weather forecasting, military, air and ground traffic control, and remote sensing. In 2017, Mao et al., developed a dual band shared-aperture circularly polarized (CP) array antenna for C- and X-band satellite communications with wide impedance bandwidths of 21% and 21.2%, respectively [13]. The metamaterial antenna also achieved high aperture efficiency, low sidelobe, and high gain. In 2018, S. Sureshkumar et al., designed a spider-like resonating structure, which resulted in a C-band and X-band microwave absorber [14]. In 2020, Abouelnaga et al., proposed a design for a 4 × 1 ultra-wide band (UWB) antenna array for C- and X-band applications [15]. In addition, Borazjani et al., developed a simple wideband negative permittivity metamaterial structure for X-band application. The metamaterial structure managed to compose a negative refractive index at a width of 3.2 GHz at the resonance frequency. Furthermore, in 2021, a compact ultra-wide band (UWB) metamaterial-inspired split-ring resonator was proposed by Jairath et al. [16]. The proposed antenna has the ability to improve the performance of the applications in WiMAX, C-band, WLAN-band, and X-band satellite communication signals by defeating the interference that occurred. Additionally, Ali et al., developed a multi-band metamaterial absorber for radar application, with a focus on C- and X-bands [17]. Although the previous studies showed benefits for microwave frequency applications, they encountered problems in terms of size. The antenna size decreases with the increase in frequency. For the C-band frequency, the size of the antenna is larger compared to the X-band and Ku-band. Consequently, the size of the metamaterial structure will continue to be the primary concern in this study.

The study of left-handed materials has been widely covered due to the unique properties that are useful for various applications. In 2017, Li et al., developed a dual band left-handed metamaterial by connecting a designed single-negative metamaterial to neighboring units with metallic wires [18]. A spherical spiral metamaterial unit cell that is smaller than λ_0_⁄60 was introduced by Smith et al., in 2018 to produce negative permeability and negative permittivity [19]. Furthermore, in 2019, H. Singh et al., designed a composite right–left hand (CRLH) metamaterial that produced a left-hand (LH) metamaterial at a frequency less than 6.8 GHz and a right-hand (RH) metamaterial at more than 6.8 GHz [20]. In the same year, He et al., proposed a double-negative metamaterial by designing a novel two-dimensional (2D) metamaterial structure with a center symmetric T-shaped unit [21]. The structure operated at K- band and Ka-band frequencies. Additionally, Faruque et al., proposed a left-handed metamaterial absorber structure that is formed by the combination of an electrical resonator and a circular ring in 2020 [22]. The developed metamaterial absorber structure has a bandwidth of 4.66 GHz and resonance frequencies in the C-, X-, and Ku-bands. In 2021, Hossain et al., introduced an electrically tunable left-handed textile-based metamaterial that consisted of a decagonal-shaped split ring resonator and a slotted ground plane integrated with RF varactor diodes [23].

The metamaterial structure is designed by the researchers to gain a variety of advantages. The design idea is based on the trial-and-error method. The type of substrate, the size of the substrate, the thickness of the substrate, the width of the metal strip, the width of the split gap, the number of the split-ring resonator, the shape of the design structure, and others are the parameters that are important to produce a metamaterial design with good performance. The parameters can be adjusted to obtain the desired electromagnetic properties. There are various shapes of metamaterial design structures that have been introduced by researchers. In 2017, Dong et al., designed double-sided spirals with a plate in the center of 64 mm × 64 mm × 0.8 mm unit cell metamaterials for wireless power transfer (WPT) system application [24]. They also demonstrated the effect of the adjustment in unit cell parameters, and it was proven that, by changing the parameters of the unit cell, there would be significant effects on resonance frequency. Additionally, in the same year, Islam et al., proposed a H-shaped unit cell metamaterial for C- and X-band applications [25]. In 2018, K. Kaur et al., developed a dual band wide angle absorber consisting of three resonators, which were two cross double-arrow shaped ones, a ring resonator with four splits at the corner, and a square-ring resonator [26]. The proposed design was applicable for energy harvesting in GSM and ISM band applications. In 2019, J. John Paul and A. Shoba Rekh proposed a dual band symmetrically E-shaped split ring resonator with a peak absorption of 99% at 5.8 GHz and 92% at 7.8 GHz [27]. Additionally, Ramachandran et al., introduced a compact left-handed circular split ring resonator (CSRR) metamaterial structure for C- and Ku-band applications [28]. In 2020, Sifat et al., proposed a single-band split-ring resonator bounded cohesive symmetric hook C-shaped metamaterial for airport surveillance radar (ASR) system [29]. In the study, a comparison of the different structures was made to differentiate the resonance and bandwidth ranges. In 2021, a triple-hexagonal SRR based on metamaterial sensor was developed for real-time applications to improve sensitivity in fuel adulteration detection [30].

The previous studies already contributed certain advantages to the applications. However, to obtain the desired performance of the metamaterial structure, there are hindrances that the researcher may face during the process, especially in the size of the structure and its electromagnetic properties. This study proposed a new combination of oval- and square-shaped split-ring resonator (SRR)-based metamaterial for satellite communications and radar applications. The metamaterial design was printed on 9 × 9 × 0.508 mm^3^ size of Rogers RO4003C substrate. The substrate was a good dielectric material and smaller in size. The metamaterial structure manifested resonance frequencies at 5.53 (C-band) and 8.81 GHz (X-band). Both resonance frequencies manifested left-handed characteristics. The effective medium ratio (EMR) of the metamaterial structure was 6.03. Thus, the metamaterial structure was compact because the EMR value was higher than the standard value, 4. The proposed design is new and applicable for the dual frequency bands. In this paper, the method used for unit cell design, simulation, measurement, and equivalent circuit are discussed in Section 2. In Section 3, the result of the simulation and experimental are compared to validate the performance of the metamaterial structure. Parametric studies such as type of substrate, the design of the metamaterial structure, and a comparison of the array structure have also been discussed further in Section 4.

## 2. Methodology

### 2.1. Unit Cell and Array Design

A few factors, such as the size of the substrate, the type of substrate, the number of resonance frequencies, and the properties of the metamaterial structure, are usually taken into account in order to produce a metamaterial structure with good performance. The unit cell and array metamaterial structures were designed and simulated using the frequency domain-based CST Microwave Studio 2018. The CST Microwave Studio software is capable of electromagnetic analysis and design at a broadband frequency. In this study, Rogers RO4003C was chosen as the substrate, and the size was 9 × 9 mm^2^. The thickness of the Rogers RO4003C substrate was 0.508 mm. The Rogers RO4003C substrate acquired a dielectric constant value of 3.55 and tangent loss (δ) of 0.0027. The metamaterial resonator was designed using copper material and was printed on the Rogers substrate. The thickness of the copper was 0.035 mm, and the electric conductivity (σ) was 5.8 × 10⁷ S/m. The full dimensions of the metamaterial structure are illustrated in Figure 1 and Table 1.

The metamaterial was designed with a combination of square- and oval-shaped split-ring resonators. There were two square-shaped split-ring resonators and three oval-shaped split-ring resonators. The first square ring was designed with a length and width of 8.00 mm, respectively, whereas the length and width of the second square ring were 6.40 mm, respectively. The gap between the first and second square ring was 0.30 mm. The square- and oval-shaped split-ring resonators were connected with a 0.50 mm wide horizontal strip that was divided in the middle by a 0.20 mm gap. Splits were placed in a parallel way at two square-shaped splits with a gap of 0.50 mm and one oval-shaped split with a gap of 0.20 mm. The three oval rings had a vertical diameter of 0.50 mm and a horizontal width of 0.3 mm. From the center, the radius of the outer oval ring was 2.5 mm, the middle oval ring was 1.8 mm, and the radius of the inner oval ring was 1.1 mm. Section 4.2 goes into greater detail about the evolution of the metamaterial design. The unit cell by itself is incompatible with the analysis. As a result, the unit cell structure has been aligned into a 4 × 4 array.

### 2.2. Numerical Simulation Method

After modelling the metamaterial structure in CST Microwave Studio 2018, a numerical simulation was performed to determine the reflection and transmission coefficients. Figure 2 depicts the wave propagation of the metamaterial structure. In this simulation, the proposed metamaterial was put in between the two ports on the positive and negative *z*-axis, respectively, as illustrated in Figure 2a. The transmitting port was represented by port 1, and the receiving port was represented by port 2. The boundary conditions for the x- and y-axes were set to perfect electric conductor (PEC) and perfect magnetic conductor (PMC), respectively. Figure 2b illustrates the boundary conditions. Because the unit cell design is smaller, the frequency domain was utilized. The characteristic impedance (Z_0_) was 50 Ω. The frequency was set from 0 GHz to 12 GHz. Once the simulation was completed, scattering parameters (S-parameters) were obtained, in which the S-parameters were composed of reflection coefficient ($S_{11}$) and transmission coefficient ($S_{21}$). The data of the S-parameters were analyzed and extracted to calculate the electromagnetic properties such as electric permittivity (ε), magnetic permeability (µ), refractive index (n), and impedance (z). The values of the electromagnetic properties were defined in Equations (1)–(7). The effective parameters of the electromagnetic properties were retrieved using the Robust method [31]. MATLAB software was used for the result retrieval.
(1)$$S_{11}=\frac{R_{01}\left(1-{\;e}^{i2{\mathrm{nk}}_{0}d}\right)}{1-{\;R}_{01}^{2}e^{i2{\mathrm{nk}}_{0}d}}$$
(2)$$S_{21}=\frac{\left(1-{\;R}_{01}^{2}\right)e^{{\mathrm{ink}}_{0}d}}{1-{\;R}_{01}^{2}e^{i2{\mathrm{nk}}_{0}d}}$$
where $R_{01}$ = z−1/z+1.
(3)$$z=\pm \;\sqrt{\frac{{\left(1+{\;S}_{11}\right)}^{2}-{\;S}_{21}^{2}}{{\left(1-{\;S}_{11}\right)}^{2}-{\;S}_{21}^{2}}}$$
(4)$$e^{{\mathrm{ink}}_{0}d}=\;Χ\;\pm i\;\sqrt{1-{\;Χ}_{2}}$$
where $Χ=1/2S_{21}(1-{\;S}_{11}^{2}+{\;S}_{21}^{2}$). The value of refractive index can be determined by Equation (4), as
(5)$$n=\frac{1}{k_{0}d}\left\{\left[\left[\ln\left(e^{{\mathrm{ink}}_{0}d}\right)\right]{}^{\prime \prime }\;+2m\pi \right]-i\left[\ln\left(e^{{\mathrm{ink}}_{0}d}\right)\right]{}^{\prime }\right\}$$

Finally, the value of ε and µ can be obtained from Equations (6) and (7),
(6)$$\varepsilon =\frac{n}{z}$$
(7)μ=nz

### 2.3. Fabrication and Experimental Method

Fabrication was carried out for experimental purposes. The oval-square-shaped split-ring resonator-based metamaterial design for 9 × 9 mm^2^ unit cell and 4 × 4 arrays was printed on Rogers RO4003C and FR-4 dielectric substrate material. The manufactured unit cell and array metamaterial structure for both substrates are depicted in Figure 3a,b. S-parameters were measured by the Vector Network Analyzer (VNA) N5227A. The fabricated metamaterial structures were placed in between waveguide ports (A-INFOMW WG) and connected to co-axial adapters 137WCAS and 112WCAS for C- and X-band, respectively. The experimental set up for the metamaterial structure is illustrated in Figure 3c.

### 2.4. Equivalent Circuit of the Oval-Square SRR

The proposed design structure of the metamaterial was made up of a series of SRR. The metamaterial can be acknowledged from the perspective of a resonant LC circuit with a resonance frequency, $f_{R}=\frac{1}{2\pi \sqrt{L_{T}}C_{T}}$, where L_T_ is the total inductance, and C_T_ is the total capacitance of the proposed SRR. A metal strip that represents effective inductance and a gap that represents effective capacitance characterize the SRR. Those elements are passive elements that will receive, store, or dissipate energy in the circuit. ADS Software was utilized to design and validate the equivalent circuit model for the proposed oval-square SRR. Figure 4 shows the circuit model of the proposed metamaterial. The outer square-shaped ring is represented by inductance and capacitance and is marked with L1, C1, L4 and C4, whereas the second square-shaped ring is represented by L2, C2, L3 and C3. Furthermore, the outer oval-shaped ring was symbolized with L13 and L16, the middle oval-shaped ring symbolized with L14 and L15, and finally, the inner oval-shaped ring is shown by L17, L18, C13 and C15. L19, L20 and C14 were assigned to the middle split horizontal strip. The passive components labelled L5 until L12 are the coupling inductors that are connected in series, whereas the components labelled C5 until C12 are the coupling capacitors that are also connected in series.

The S_21_ results from the simulation and the equivalent circuit are compared in Figure 5. The resonance frequencies for the equivalent circuit model are given at 5.53 GHz and 8.70 GHz with magnitudes of −47.08 dB and −29.40 dB, respectively. Meanwhile, the resonance frequencies of S_21_ for the simulation results are 5.52 GHz and 8.81 GHz with magnitudes of −43.03 dB and −17.19 dB, respectively. Because the equivalent circuit is made up of a series of passive elements, the energy dissipation is customized by the inductance and capacitance, the magnitude of S_21_ tends to be smaller. Despite having a difference in the magnitude of the S_21_, the resonance frequencies of the simulation and equivalent circuit results are almost the same.

## 3. Results and Discussion

Unit cell and array metamaterial analyses are discussed in this section. Parametric studies, such as substrate type and metamaterial structure design, are also presented in order to compare and identify the optimal performance of the metamaterial structure. The effective medium parameters and S-parameter results from simulation and experiment are thoroughly discussed to determine whether the proposed oval-square-shaped SRR metamaterial-based structure can be classified as having left-handed metamaterial properties suitable for satellite and radar applications.

### 3.1. Unit Cell Metamaterial Structure Analysis

The numerical and experimental results have been analyzed accordingly. S-parameters such as reflection coefficient (S_11_) and transmission coefficient (S_21_) were retrieved from the CST Microwave Studio software and plotted as shown in Figure 6a. According to the simulation, the oval-square-shaped SRR structure displayed S_11_ with magnitudes of −34.64 and −24.66 dB at resonance frequencies of 7.33 and 9.06 GHz. According to the S_21_ result, the metamaterial design had a dual frequency band. The resonance frequencies were 5.52 GHz and 8.81 GHz, with magnitudes of −43.03 dB and −17.19 dB, respectively. The experimental results for S_21_, as shown in Figure 6a, revealed that the proposed design had dual frequency bands at 5.53 GHz and 8.31 GHz, with magnitudes of −31.18 and −17.90 dB, respectively. Meanwhile, the S_11_ result presented resonance frequencies at 7.51 GHz and 8.31 GHz with magnitudes of −17.76 dB and −17.90 dB, respectively. The C-band resonance frequency of S_21_ has a percentage difference of 0.18%, whereas the X-band resonance frequency difference was 5.68%. It was discovered that the simulated and experimental findings were practically identical. This is because the results can be influenced by several factors, which allow the discrepancies to happen between both methods. The measurement method was utilized to validate the simulation data, and there might be a calibration error of the Agilent N5227A vector network analyzer by utilizing the Agilent N4694-60001 Ecal device. Moreover, there has always been some mutual resonance effect between the transmitting and receiving ends of the waveguide. The array prototype of the metamaterial was measured by the pyramidal horn antenna. Therefore, the possible reason for the discrepancies in results might be due to the management of two horn antennas for the metamaterial array cell measurement. On the other hand, there might be errors that occurred during the fabrication process. Finally, the permittivity of the substrate is an important factor in the variation of simulated and measured results. The resonance frequency depends on the substrate material’s permittivity. If the permittivity is increased, then the resonant peaks are shifted toward the lower frequency. When the dielectric constant is increased, each capacitance value between the ground and the radiating elements is also raised. Since each capacitance of the radiating elements is in series with others, the system’s equivalent capacitance is reduced. The decrease in the equivalent capacitance is another reason for the variation in the results. On a side note, the metamaterial structure’s numerical simulation can be extended to 18 GHz because it can accomplish Ku-band due to the design’s employment of a number of SRR. However, due to limits in the software’s functionality, we primarily concentrated on the frequency range up to 12 GHz in this study.

The method of retrieval of the effective medium parameter was required to characterize the electromagnetic properties of artificial materials. The primary electromagnetic properties that needed to be studied in order to classify metamaterials were electric permittivity and magnetic permeability. The values of impedance and refractive index were extracted from the scattering parameter, S_11_ and S_21_ data, and subsequently, ε and µ were also retrieved. The results can be seen in Figure 6b, where the real part values of relative ε and µ are below zero at a specified frequency range. From 5.94 GHz to 12 GHz, the value of relative ε was negative, with the maximum value of −518.34 at the frequency of 8.44 GHz. Permittivity refers to a material’s ability to store electrical energy in the presence of an electric field. When the value of ε is low, it may be due to the lower tendency of electric polarization. The result of relative µ shows that the frequency exhibited a negative value from 0 GHz, with three maximum values of −513.23, −550.36 and −92.57 at frequencies of 5.53, 6.05 and 8.92 GHz, respectively. Furthermore, the refractive index’s real part was negative. Thus, from the simulated result, the proposed oval-square-shaped SRR displayed left-handed metamaterial properties. Additionally, the real part of the impedance value is above zero, as shown in Figure 6c. At frequencies of 5.21, 5.99 and 8.92 GHz, there were three impedance peaks with values of 54.36, 12.41, and 1.82 Ω, respectively. The suggested structure is a passive medium if the impedance value is greater than zero.

### 3.2. The Analysis of Electromagnetic Field in Metamaterial

Metamaterials, in general, are engineered structures that are occasionally arranged to exhibit intriguing electromagnetic properties not seen in conventional materials. The interaction of the metamaterial with electromagnetic waves can be described by Maxwell’s equations [32].

The structure of artificial materials can be modified to obtain appropriate values of permittivity and permeability for applications. Figure 7 shows the electromagnetic properties of the artificial material. Electromagnetic waves are associated with a frequency and a wavelength, and they travel at the speed of light, c. The relationship between the wave characteristics can be explained by the equation c = fλ. The higher the frequency, the shorter the wavelength. As a result, a higher frequency carries more energy. That being the case, electromagnetic waves with a frequency of 8.81 GHz interact more strongly than electromagnetic waves at the frequency of 5.52 GHz. The red zone at the metamaterial structure of both resonance frequencies in Figure 7a indicates that there was a strong electric field reaction at the gap of the square shaped SRR and at the gap of the center of the oval-shaped metal. It means that the capacitor stores and releases charge across the SRR gaps. Meanwhile, the magnetic field result shown in Figure 7b indicated that the metal strip acted as an inductor, storing electrical energy in the form of magnetic energy. Because the magnetic field was created by an electric current, looking at Figure 7b,c, it it clear that the magnetic field was dependent on the current density. Because there was low current flow at a resonance frequency of 5.52 GHz, the magnetic field at the metal gap was minimal. Due to the larger current density, a strong magnetic field was sensed at the metal strip that joined the square and oval shapes. Furthermore, the current density was lower at the metal gap and the center of the oval-shaped SRR at the resonance frequency of 8.81 GHz, affecting the weaker magnetic field interaction. Due to the high concentration of current density, the magnetic field was stronger at the metal strip of the square and outer oval shapes of the SRR. Figure 7c demonstrates the proposed SRR’s current distribution. As can be observed, the current was concentrated throughout the oval-square shape SRR. At the resonance frequency of 8.81 GHz, large amounts of current were concentrated practically all across the oval-square-shaped SRR, as opposed to the resonance frequency of 5.52 GHz, where the current was primarily concentrated at the connection metal between the oval- and square-shaped SRRs. A certain part of the metal cannot be coupled with the incident waves because the electric field polarization direction of the incident wave was in the *y*-axis direction; meanwhile, the magnetic field polarization direction was in the *x*-axis direction. Furthermore, the oval-square-shaped SRR’s symmetrical design allows the current to flow in the opposite direction.

On the other hand, the compactness of the metamaterial can be explained by the effective medium ratio (EMR), which can be described by the ratio of wavelengths to the size of the unit cell.
(8)$$\mathrm{EMR}=\;\mathrm{Wavelength}\;\left(\lambda \right)\;/\mathrm{Unit}\;\mathrm{cell}\;\mathrm{length}\;\left(L\right)$$

The EMR of the metamaterial should be higher than the standard value of EMR, which is 4. The EMR of the proposed oval-square-shaped SRR was apparently 6.04, which meets the standard value.

### 3.3. Array Metamaterial Structure

The array structure of metamaterial was simulated several times with different dimensions, such as 2 × 2, 3 × 3 and 4 × 4, in which the width and length of the structures are 18 × 18 mm^2^, 27 × 27 mm^2^, and 36 × 36 mm^2^, respectively. Figure 8 depicts the outcome of $S_{11}$ and $S_{21}$ of the array structure. The magnitude of $S_{21}$ for the 2 × 2 array structure was simulated to be at the frequencies of 5.52 and 8.81 GHz with magnitudes of −41.54 and −17.16 dB, respectively. The resonance frequency was similar to the resonance frequency of the unit cell structure, despite having a 0.03 to 1.49 dB difference in the magnitude of $S_{21}.$ For 3 × 3 array structure, the magnitude of $S_{21}$ was −41.34 and −17.24 dB at the frequencies of 5.52 and 8.80 GHz. The resonance frequency for X-band had a difference of 0.01 GHz compared to the unit cell structure and 0.05 to 1.69 dB difference in the magnitude of $S_{21}$. Meanwhile, for the 4 × 4 array structure, the magnitude of $S_{21}$ was −40.89 and −17.23 dB at the frequencies of 5.51 and 8.80 GHz. When compared to the unit cell structure, the resonance frequency for both frequency bands differed by 0.01 GHz, and the magnitude of $S_{21}$ differed by 0.04 to 2.14 dB. According to Figure 6, the s-parameters of the array structures were nearly identical, with only a small difference in the value of the resonance frequency and its magnitude of $S_{21}.$ As a result, the array structure in this proposed design can be any size to validate the unit cell metamaterial structure and for satellite and radar applications.

As the size of the metamaterial structure is the primary focus of this research, a smaller array of metamaterial can be used for the application. However, in this study, a larger array structure was chosen to validate with a unit cell metamaterial structure because errors during the experiment can be minimized, and it has a low $S_{21}$ loss. Figure 9 compares the S-parameters for unit cell and array metamaterial structures. The magnitudes of $S_{21}$ for unit cell metamaterial structure were −43.03 and −17.19 dB at the frequencies of 5.52 and 8.81 GHz, whereas the magnitudes of $S_{21}$ of the 4 × 4 array structure were −40.89 and −17.23 dB at the frequencies of 5.51 and 8.80 GHz. The magnitude of $S_{21}$ for array structure is lower compared to the unit cell structure.

## 4. Parametric Study of the Oval-Square-Shaped SRR Structure

### 4.1. Different Type of the Substrate

Every dielectric substrate material has different properties. In this study, the metamaterial structure was simulated using several substrate materials such as Flame Retardant 4 (FR-4), Rogers RO4003C, Rogers RT5880, and Rogers RT5870. The value of thickness, dielectric constant, and tangent loss of the substrates are different. The properties of the substrate materials were as shown in Table 2.

The simulated result was plotted as illustrated in Figure 10. The FR-4 substrate showed that it only exhibited C-band with a frequency ranged between 4.09 GHz and 5.12 GHz, and the magnitude of $S_{21}$ was −29.38 dB. Meanwhile, the Rogers RO4003C substrate produced C- and X-band frequencies ranging from 4.98 GHz to 5.92 GHz and 8.76 GHz to 8.86 GHz, respectively. The magnitudes of $S_{21}$ were −43.03 and −17.19 dB. C- and X-band frequencies ranged between 5.4 GHz to 6.42 GHz and 9.98 GHz to 10.07 GHz, 5.27 GHz to 6.28 GHz, and 9.72 GHz to 9.80 GHz, respectively, on Rogers RT5880 and Rogers RT5870 substrates. The magnitudes of $S_{21}$ for Rogers RT5880 were −44.05 and −18.35 dB, whereas for Rogers RT5870, they were −42.76 and −18.15 dB. A comparison of the S-parameter results for both substrates is shown in Table 3. All the substrates manifested dual frequency bands except FR-4, which manifested only a single resonance frequency. Generally, the dielectric constant is defined by the ratio of the electric permittivity of the material to the electric permittivity of free space, such as vacuum. The dielectric constant can also be called the electric permittivity. From the properties of the substrates, it can be understood that FR-4 has the highest dielectric constant. Hence, the shifting in the value of resonance frequency occurred because of the different values of the dielectric constant of the substrate. Overall, from the graph, it can be observed that the resonance frequency increases with the lower value of the dielectric constant. It can also be concluded that a low dielectric constant can be used to achieve a higher resonance frequency. Rogers RO4003C has low loss, and the dielectric constant is higher when compared to Rogers RT5880 and Rogers RT5870. It also exhibits dual frequency bands when compared to FR-4, so it was chosen for further investigation.

### 4.2. The Design of Metamaterial Structure

The metamaterial structure was altered until the desired simulated results were achieved. In this study, the metamaterial structure was designed for C- and X-band applications. Thus, the design was tweaked in a series of steps, as shown in Figure 11. Firstly, the structure just included three rings of oval-shaped SRR. The simulated result, on the other hand, revealed that the metamaterial structure obtained a resonance frequency of 13.34 GHz (Ku-band). Then, a square-shaped resonator was added. The transmission coefficient result improved to dual resonance frequencies of 7.52 GHz (C-band) and 15.08 GHz (Ku-band). Since this study was focused on the C- and X-bands, the design 3 structure was created by adding another square-shaped split ring resonator. The resonance frequency was gained at 7.02 GHz (C-band), 10.30 GHz (X-band), and 15.10 GHz (Ku-band). The metamaterial structure must have a compact size; therefore, a lower resonance frequency needs to be achieved by the design. Having said that, another metamaterial structure was designed by connecting each of the rings with a split horizontal bar. However, the resonance frequency manifested by design 3 shifted to only a single frequency at 14.56 GHz (Ku-band). Finally, the metamaterial structure was designed with a gap between the oval and square-shaped rings to split the ring. The resonance frequency was improved to 5.52 GHz (C-band) and 8.81 GHz (X-band). The result of the $S_{21}$ for the performance of the steps taken was assembled in Figure 12.

From the simulated result, it can be identified that the design of the metamaterial structure affected the resonance frequency. In further observations, the oval-square-shaped SRR was rotated 90° to the right, as shown in Figure 13. It can be seen that, from the figure, the S-parameter result of design A was different from that of design B. Based on the simulated result, design A exhibited dual band resonance frequencies at 7.21 GHz (C-band) and 10.72 GHz (X-band) with magnitudes of −42.05 and −40.24 dB, respectively. Meanwhile, design B also manifested dual bands at the resonance frequency of 5.52 GHz (C-band) with a magnitude of −43.03 dB, and 8.81 GHz (X-band) with a magnitude of −17.19 dB. Based on the result, although the design is identical, by changing the position of the metal gap, it will affect the resonance frequency. The difference in the resonance frequencies happened due to the electric and magnetic field polarization directions. Hence, when designing the metamaterial structure, researchers can consider rotating the position of the metal to identify the best structure for the desired properties for the applications.

## 5. Comparison with the Previous Studies

The comparison between the proposed design and the previous studies was shown in Table 4. Metamaterial enables the extreme miniaturization of existing devices and applications, as well as the customization of novel characteristics not available in current satellite and radar applications. Higher resonance frequencies are typically easier to produce when using a smaller metamaterial structure. As stated in Table 4, the previous studies used a larger size of unit cell and an array metamaterial structure. For example, Islam et al. [33] used 10 × 10 mm^2^ size of FR-4 unit cell, which resulted in only a single frequency band, and the characteristic of the proposed design is ENZ. The same goes for Hasan et al. [34] and Faruque et al. [35]; they proposed double-negative metamaterial structures for a single frequency band, which is X-band. Besides the size of the metamaterial structure, other parameters such as the design of the metamaterial must be considered to produce the desired novel properties. The proposed design consists of five rings of resonators, resulting in a dual-frequency band. Hossain et al. [36] designed a double T-U-shaped metamaterial, which resulted in triple-frequency bands. Thus, it was shown that a multiple number of ring resonators will result in multiple resonance frequencies. Furthermore, the proposed design also exhibited the ideal properties of the artificial material, which is a left-handed metamaterial. When compared to other studies, they only managed to produce ENZ and DNG metamaterials. The proposed metamaterial also uses different types of substrate. Even though the dielectric constant of Rogers RO4003C is lower than that of FR-4 and nickel aluminate, the proposed metamaterial achieved better results compared to the previous studies. With the highlighted performance of the novel metamaterial structure, it can be deduced that the designation of compact oval-square-shaped SRR using Rogers RO4003C substrate is suitable for current satellite and radar applications. On a side note, the proposed metamaterial also needed to be improved in terms of design structure. Although it may provide some benefits to the targeted applications, the complicated design of the proposed SRR may result in additional work for the researchers.

## 6. Conclusions

A novel left-handed metamaterial was demonstrated using a novel oval-square-shaped split-ring resonator. The proposed metamaterial structure achieved resonance frequencies of 5.52 (C-band) and 8.81 GHz (X-band). With a lower percentage difference for resonance frequencies of S_21_, the experimental results were correlated to the simulated results. The simulation and equivalent circuit results are also almost similar. The EMR for the metamaterial design structure of the Rogers 4003C substrate is 6.03, demonstrating the proposed design’s compactness. In this study, the size of the proposed metamaterial structure was minimized while it exhibited the ideal metamaterial properties. With the proposed metamaterial structure’s leading advantages, it is suitable for enhancing emerging satellite communications and radar applications.

## Figures and Tables

**Figure 1 micromachines-13-00578-f001:**
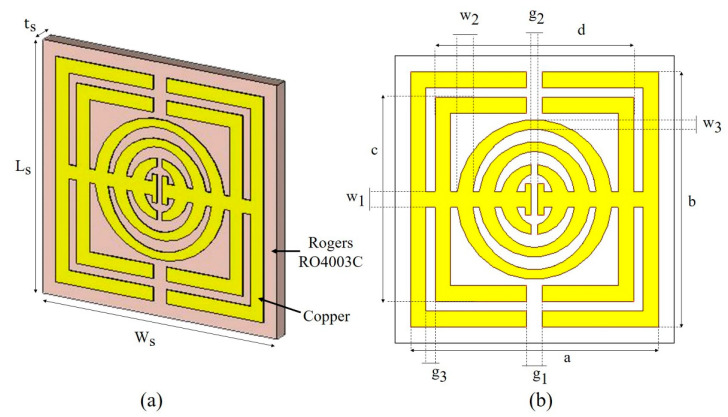
Unit cell metamaterial design, (**a**) perspective view and (**b**) top view of unit cell metamaterial design in CST Microwave Studio.

**Figure 2 micromachines-13-00578-f002:**
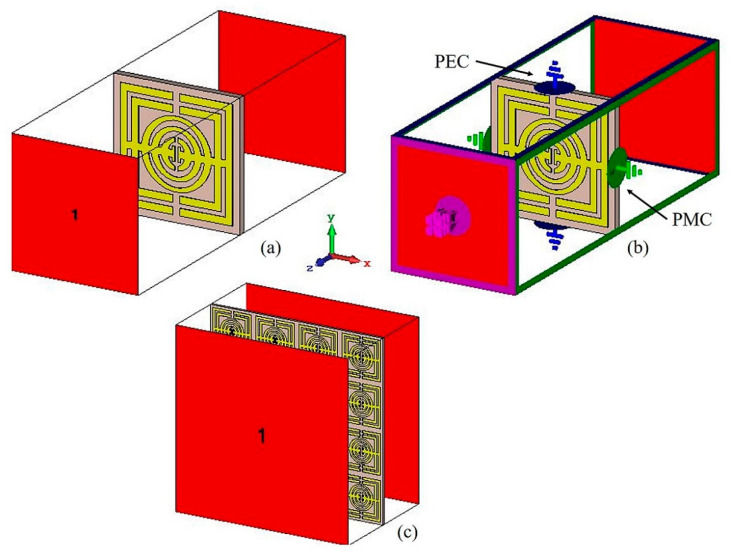
Wave propagation simulation, (**a**) unit cell metamaterial, (**b**) boundary conditions, and (**c**) 4 × 4 arrays metamaterial.

**Figure 3 micromachines-13-00578-f003:**
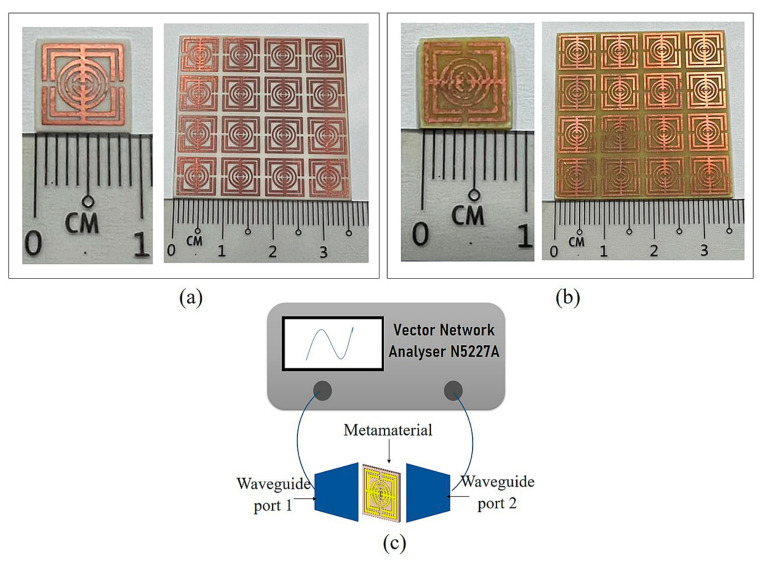
Fabricated unit cell and arrays metamaterial, (**a**) Rogers RO4003C, (**b**) FR-4 and the illustration of experimental set up, (**c**) VNA.

**Figure 4 micromachines-13-00578-f004:**
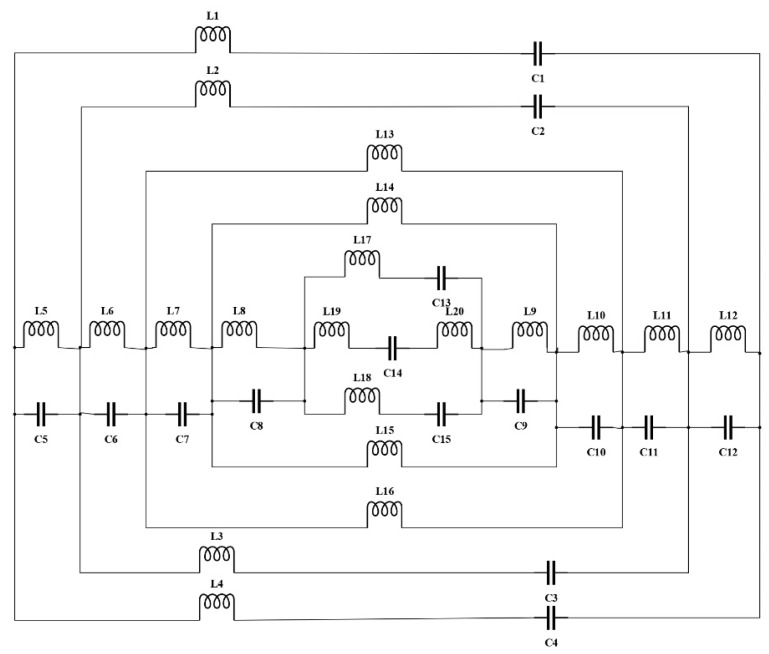
The equivalent circuit model of the proposed metamaterial.

**Figure 5 micromachines-13-00578-f005:**
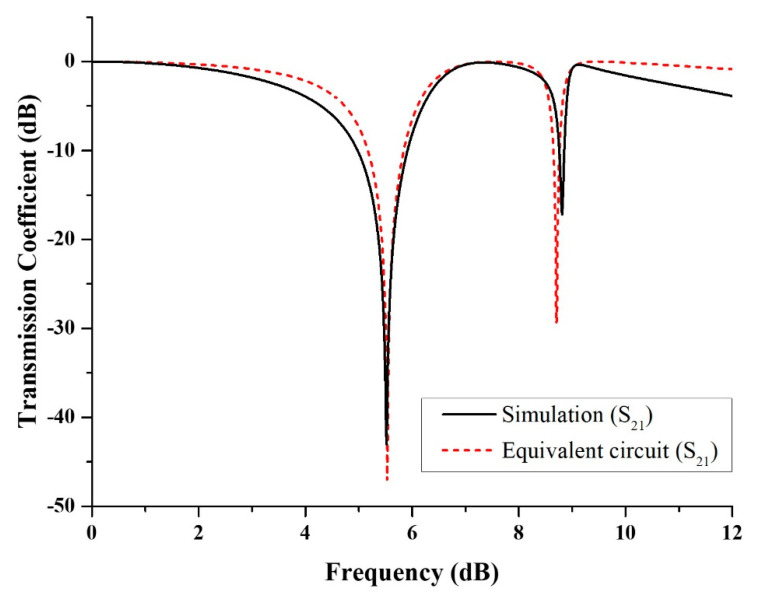
The comparison of S_21_ results for simulation and equivalent circuit.

**Figure 6 micromachines-13-00578-f006:**
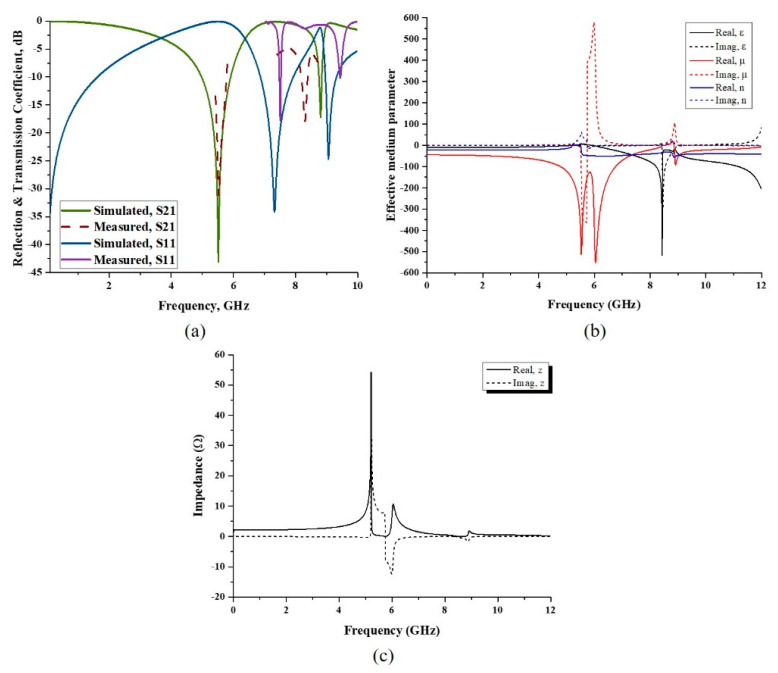
(**a**) Transmission (S_21_) and reflection coefficient (S_11_) of simulation and measured result, (**b**) simulation result for permittivity, permeability, and refractive index, and (**c**) impedance.

**Figure 7 micromachines-13-00578-f007:**
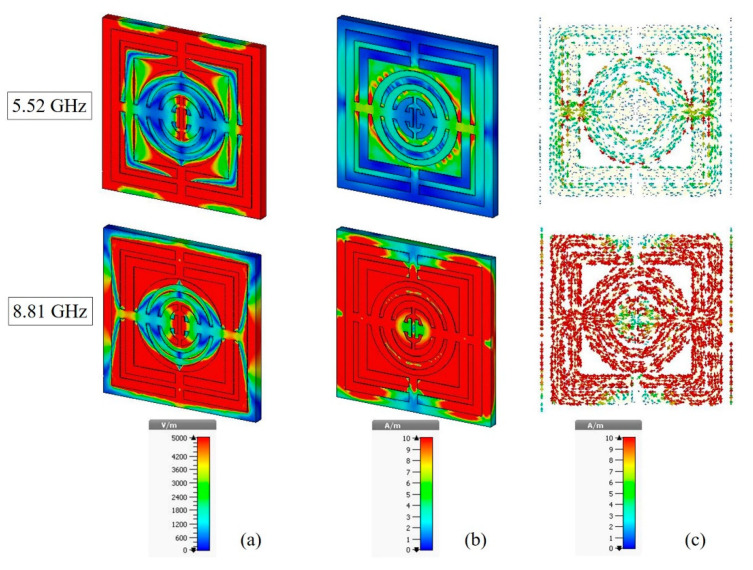
(**a**) Electric field, (**b**) magnetic field, and (**c**) surface current distribution.

**Figure 8 micromachines-13-00578-f008:**
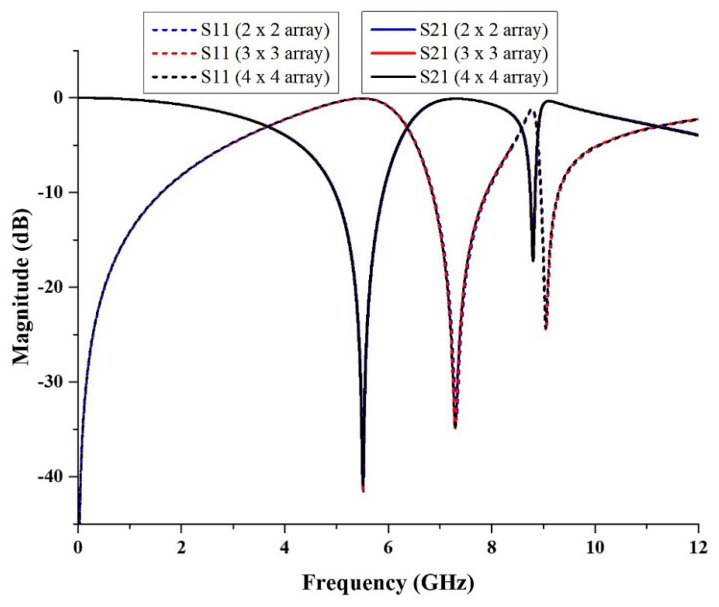
The comparison of the S-parameter results for 2 × 2, 3 × 3 and 4 × 4 arrays.

**Figure 9 micromachines-13-00578-f009:**
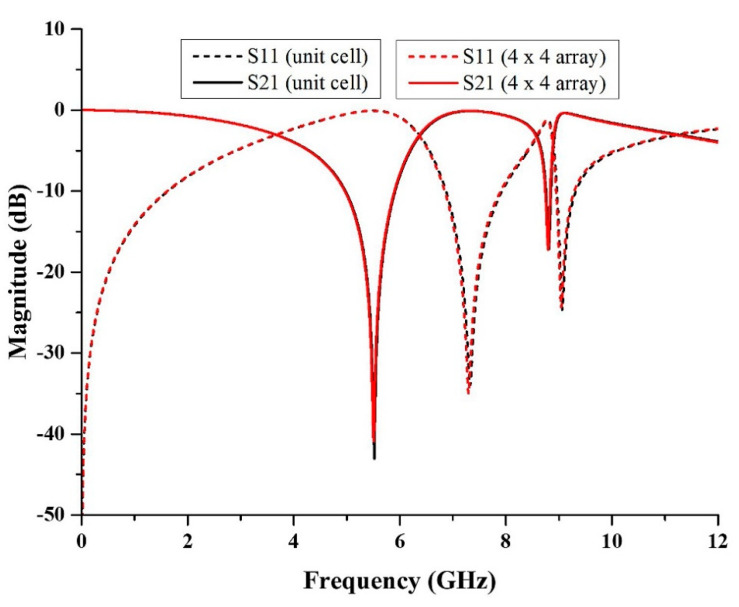
Comparison of the S-parameters for the unit cell metamaterial structure and 4 × 4 array structure.

**Figure 10 micromachines-13-00578-f010:**
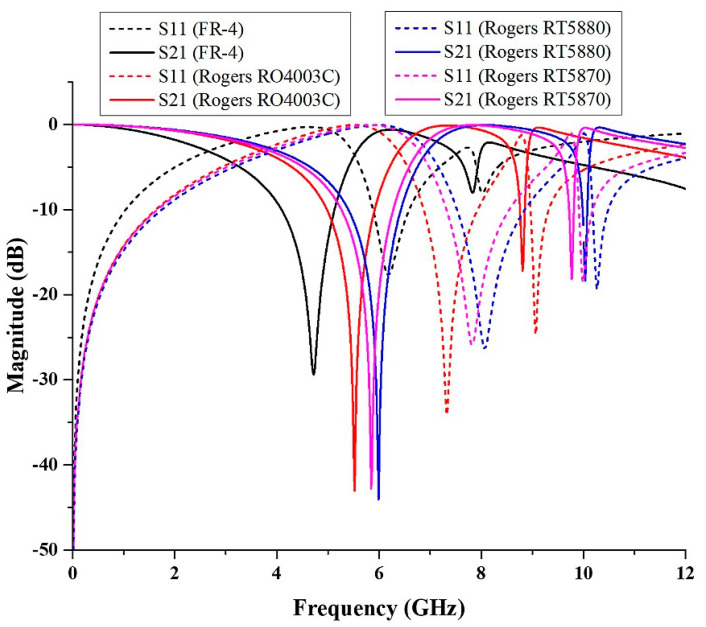
The comparison of $S_{11}$ and $S_{21}$ for the different type of substrate materials.

**Figure 11 micromachines-13-00578-f011:**
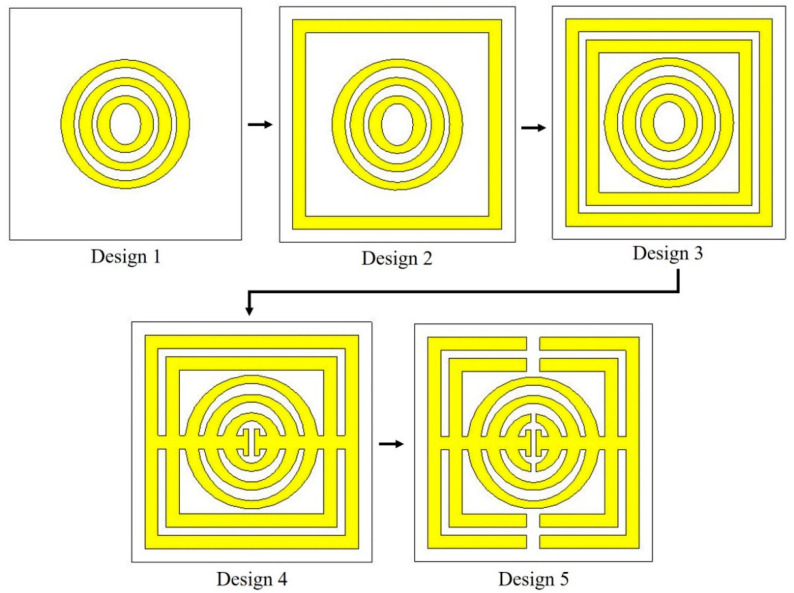
The steps of developing the new oval-square-shaped metamaterial structure.

**Figure 12 micromachines-13-00578-f012:**
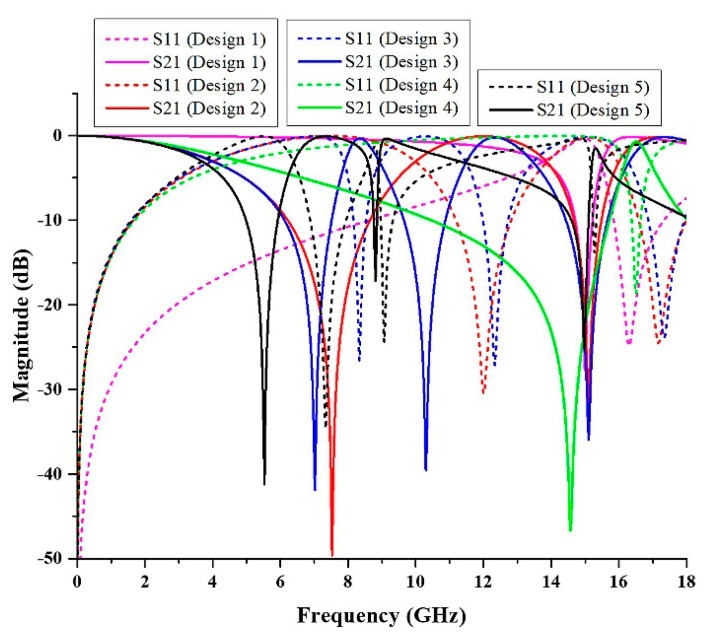
The comparison for the S-parameters of the different design of metamaterials.

**Figure 13 micromachines-13-00578-f013:**
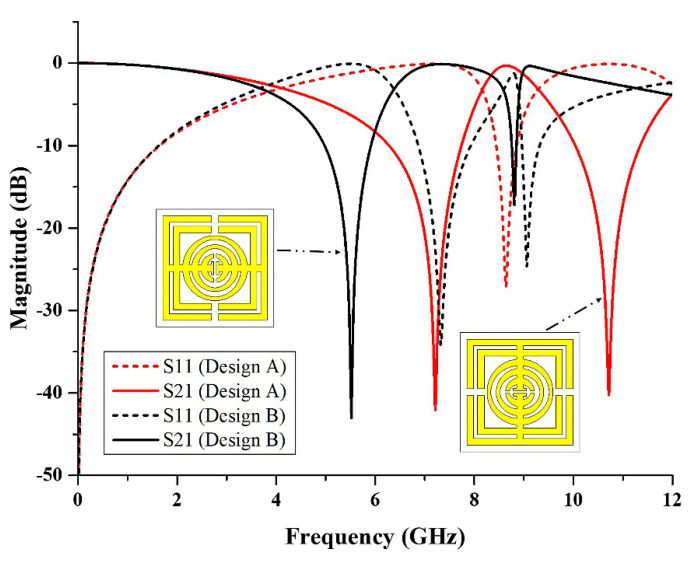
The comparison of S-parameters for design A and design B.

**Table 1 micromachines-13-00578-t001:** Parameters of the oval-square-shaped split-ring resonator-based metamaterial.

Parameters	Dimensions, mm
$\mathrm{Thickness}\;\mathrm{of}\;\mathrm{substrate},\;t_{s}$	0.508
$L_{s}$ and $W_{s}$	9.00
a and b	8.00
c and d	6.40
$w_{1}$, $w_{2}$ and $g_{1}$	0.50
$w_{3}$ and $g_{3}$	0.30
$g_{2}$	0.20

**Table 2 micromachines-13-00578-t002:** The properties of the substrate materials.

Substrate	Thickness	Dielectric Constant	Tangent Loss
FR-4	1.6	4.3	0.025
Rogers RO4003C	0.508	3.55	0.027
Rogers RT5870	1.575	2.33	0.0012
Rogers RT5880	1.575	2.2	0.0009

**Table 3 micromachines-13-00578-t003:** Comparison of the S-parameters result for metamaterial structure using different substrates.

Substrate	Frequency Band	$\mathrm{Resonance}\;\mathrm{Frequency}\;\mathrm{of}\;S_{21},\;\mathrm{GHz}$	Frequency Range, GHz	Magnitude, dB
FR-4	C-band	4.72	4.09–5.12	−29.38
Rogers RO4003C	C- and X-band	5.52, 8.81	4.98–5.92, 8.76–8.86	−43.03, −17.19
Rogers RT5870	C- and X-band	5.84, 9.77	5.27–6.28, 9.72–9.80	−42.76, −18.15
Rogers RT5880	C- and X-band	5.99, 10.03	5.4–6.42, 9.98–10.07	−44.05, −18.35

**Table 4 micromachines-13-00578-t004:** The comparison between the proposed design and the previous studies.

References	Dimension of Unit Cell Structure (mm^2^)	Dimension of Array Structure (mm^2^)	Frequency Band (GHz)	Characteristic	Type of Substrate
Islam et al. [33]	10 × 10	-	C-band	Epsilon-near-zero (ENZ)	FR-4
Hossain et al. [36]	10.5 × 11	55 × 42.5	L, C and Ku-bands	DNG	FR-4
Hasan et al. [34]	10 × 10	-	X-band	DNG	FR-4
Faruque et al. [35]	12.5 × 10	25 × 20	X-band	DNG	Nickel aluminate
Proposed design	9 × 9	36 × 36	C and X-bands	Left-handed	Rogers RO4003C

## Data Availability

All the data are available within the manuscript.

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
