# Peer review of "An Oval-Square Shaped Split Ring Resonator Based Left-Handed Metamaterial for Satellite Communications and Radar Applications"

_micromachines, 2022, doi:10.3390/mi13040578_

Round 1

Reviewer 1 Report

Hello,authors proposed a Left-handed metamaterial using a novel oval-square shaped split 457 ring resonators. My comments are as below:1-measurement results haven't been presented clearly. S11 should be added to the manuscript.2-what's the main reason for the difference between simulation and measured results? 3- Would you please clearly demonstrate the advantages of the proposed structure versus other references?4-References and literature review can be improved.5- Would you please add a brief design methodology for the proposed design. 6- Is it possible to extract an equivalent circuit for the proposed structo

Author Response

As attached.

Reviewer 2 Report

It seems to me as a cut-and-try design of a metamaterial units without no serious scientific insight. Number of similar structures exist already in literature....

- The material is simulated and measured, but the results are not compared.

- I am also curious why eps/mu and impedance in Fig 4 c,d is given in dB??

- Line 264 you say that epsilon is -518 dB? Such extremely low value? I have concerns about scientific soundness then..

- Maxwell equations are well known, (8-13) is not neccessary.

- Overall the paper is very long and quite hard to follow.

- English language should be improved, e.g.there is striking number of "was", "were" words

Author Response

As attached.

Reviewer 3 Report

Title: A New Oval-Square Shaped Split Ring Resonator Based Left-handed Metamaterial for Satellite Communications and Radar Applications

Authors: Ismatul Nisak Idrus, Mohammad Rashed Iqbal Faruque, Sabirin Abdullah, Mayeen Uddin Khandaker, Abdelmoneim Sulieman and Nissren Tamam

Manuscript ID: micromachines-1599839

Recommendation: Publish after minor revision

Comments: This manuscript by Idrus et al. designed and tested a split ring resonator combining the oval and square shape with left-handed metamaterial properties. Rogers RO4003C was used as the substrate and five shape designs were tested. The resonance frequency of the resonator of different shape, size, and substrate was simulated and compared experimentally using Computer Simulation Technique (CST) Microwave Studio. Both results showed that the designed resonator was suitable for C-band and X-band applications. Due to the high quality of the investigation and the care with which this manuscript was prepared there is little ground for review. Therefore, I recommend publish on Micromachines after addressing below points:

  1. Delete “new” in the title. One should let the work itself decide whether it is a first time finding or not.
  2. Page 1, Line 28, delete “ago”.
  3. Page 2, Line 46, delete “was”; Line 49, change to “exhibit”; Line 55, delete “are”.
  4. Page 2, Line 58, check the redundancy of “… MNG (permeability negative) when the permeability is negative…”.
  5. Page 3, Lines 143-147, the authors should provide “section number” in the corresponding descriptions. E.g., “the method used for unit cell design, simulation and measurement were discussed IN SECTION II”.
  6. In Figure 3, panel (b) is the same as panel (a). There is no experimental set up scheme in Figure 3b.
  7. Figure 4a caption, change to “(a) Simulation result of transmission (S11) and reflection coefficient (S21)”.
  8. Page 8, line 242, change to “showed”.
  9. Page 9, line 288, change “split ring resonator (SRR).” to “SRR”.
  10. In Figure 5, the unit of the intensity bar for (b) magnetic field and (c) surface current distribution is miss labelled.
  11. In Figure 9, change the double arrow in design 3 ↔ 4 to only forward arrow.

Author Response

As attached.

Round 2

Reviewer 2 Report

Thanks for your reply.

I still don't understand the units for Eps and Mu in Fig. 6b "Amplitude"? If its relative mu/eps, it has no dimension.. 

Are the authors aware that -550 dB is a number of order 10^-55 ? This is the first paper ever where someone uses such small number :) Anyway, something is wrong here. Even -100 dB is below any physical reality. Probably the other results are correct, but this is not sounding well.

Author Response

As  attached.

Round 3

Reviewer 2 Report

I still have doubts about the extremely low permittivity, but anyway, authors spent some effort to explain this.